# On the Zeros of the Differential Polynomials $\varphi f^l (f^{(k)})^n - a$

**Jiantang Lu and Junfeng Xu ***

School of Mathematics, Wuyi University, Jiangmen 529000, China; lujt_math@163.com
* Correspondence: jfxu@wyu.edu.cn

**Abstract:** Letting $f$ be a transcendental meromorphic function, we consider the value distribution of the differential polynomials $\varphi f^l (f^{(k)})^n - a$, where $\varphi(\not\equiv 0)$ is a small function of $f$, $l(\geq 2)$, $n(\geq 1)$, $k(\geq 1)$ are integers and $a$ is a non-zero constant, and obtain an important inequality concerning the reduced counting function of $\varphi f^l (f^{(k)})^n - a$. Our results improve and generalize the results obtained by Xu and Ye, Karmakar and Sahoo, Chakraborty et.al., and Chen and Huang.

**Keywords:** meromorphic functions; differential polynomials; value distribution; small functions

**MSC:** 30D35; 30D05

## 1. Introduction and Results

In this paper, we assumed that the reader is familiar with the notations of the Nevanlinna theory (see, e.g., [1,2]).Let $f(z)$ and $\alpha(z)$ be two meromorphic functions in the complex plane. If $T(r, \alpha) = S(r, f)$, then $\alpha(z)$ is called a small function of $f(z)$.

In 1959, W.K. Hayman first considered the value distribution of differential polynomials in his seminal paper, and proved that if $f$ is a transcendental meromorphic function and $l(\geq 3)$ is an integer, then $\Psi = f^l f' - a$ has infinitely many zeros for a finite non-zero complex value $a$ (see [3]). Moreover, Hayman conjectured that the conclusion remains valid for the cases $l = 1, 2$ ([4]). In 1979, Mues [5] confirmed the case $l = 2$, and Bergweiler and Eremenko [6], Chen and Fang [7] proved the case $l = 1$ in 1995. Since then, there was a lot of research on the value distribution of differential polynomials. Sons [8] and Hennekemper [9] generalized Hayman's result, and obtained the value distribution of $\Psi = f^n (f')^{n_1} \cdots (f^{(k)}) - a$ for $n \geq 2, k \geq 1$ and $(f^{n+k})^{(k)} - a$ for $n > 2, k \geq 1$, respectively. Zhang [10] also investigated the simple differential polynomial $f^2 f' - 1$ and give a precise inequality $T(r, f) \leq 6N(r, 1; f^2 f') + S(r, f)$. Huang and Gu [11] generalized the result by using $f^{(k)}$ instead of $f'$.

From the above, we know that the results on the zeros of differential polynomials have three forms. The first is purely qualitative: for example, $\Psi$ has infinitely many zeros. The second is the "semi"-quantitative: for example, $\limsup \overline{N}(r, 0; \Psi)/T(r, f) > 0$ (see [12]). The third is quantitative; that is, the characteristic function estimated by a counting function (or reduced counting function).

It is natural to consider the characteristic function estimated by a reduced counting function for the results of Zhang [10], and Huang and Gu [11]. In 2011, J. F. Xu et al. [13] proved the inequality $T(r, f) \leq M\overline{N}(r, 1; f^2 f^{(k)}) + S(r, f)$, where $M = 6$ except for $k = 2$, $M = 10$. Later, Karmakar and Sahoo [14] found the coefficient is also 6 when $k = 2$ for the inequality. Moreover, they improved the result of Xu et al., and obtained a unified inequality for $l(\geq 2), k(\geq 1)$. That is, if $f$ is a transcendental meromorphic function, and $l(\geq 2), k(\geq 1)$ are any integers, then $T(r, f) \leq \frac{6}{2l-3} \overline{N}(r, 1; f^l f^{(k)}) + S(r, f)$.

Another question is whether or not the differential polynomial takes the small function infinite times. This is a difficult question. Xu and Yi [15] gave a precise inequality for $\varphi f^2 f' - 1$, and proved $T(r, f) < 6\overline{N}(r, 1; f^2 f') + S(r, f)$, which also generalized a result of Q.D. Zhang [16] that proved the inequality by the counting function. Recently, Xu and

Ye [17] obtained an inequality that $T(r, f) < 6\overline{N}(r, 1; \varphi f^2 f'^2) + S(r, f)$. Also, Chakraborty, Saha and Pal [18] extended the inequality by replacing $f^{(k)}$ by $(f^{(k)})^n$, but there is the restriction on $f$ with no simple pole, where $l(\geq 2), n(\geq 1), k(\geq 1)$ are integers and obtained $T(r, f) \leq \frac{6}{2l-3}\overline{N}(r, 1; f^l(f^{(k)})^n) + S(r, f)$. Later, Chen and Huang refined the coefficient of the result of Chakraborty, Saha and Pal, but there is another restriction of $f$ with finite order.

**Theorem 1** ([19])**.** *Let $f$ be a transcendental meromorphic function with finite order in the complex plane, $l(\geq 2), n(\geq 1), k(\geq 1)$ be integers and $a$ be a non-zero constant. Then,*

$$T(r, f) < M\overline{N}(r, a : f^l(f^{(k)})^n) + S(r, f),$$

*where $M = \min\{\dfrac{1}{l-2}, 6\}$.*

A natural question is raised as to whether the above inequality still holds if one gets rid of some restrictions on $f$. Moreover, the constant $a$ is replaced by a small function of $f$. Now, we consider the characteristic function estimate of more general forms $\varphi f^l(f^{(k)})^n - a$ for a non-zero constant $a$, integers $l \geq 2, n \geq 1$, and $k \geq 1$, and obtain its quantitative result as follows:

**Theorem 2.** *Let $f$ be a transcendental meromorphic function and $\varphi(\not\equiv 0)$ be a small function of $f$, $l(\geq 2), n(\geq 1), k(\geq 1)$ be integers, and $a$ be a non-zero constant. Then,*

$$T(r, f) \leq M\overline{N}(r, a; \varphi f^l(f^{(k)})^n) + S(r, f),$$

*where $M = 5$ if $l = 2$ and $M = \dfrac{1}{l-2}$ if $l \geq 3$.*

**Remark 1.** *Obviously, Theorem 2 improves the results of Xu et al. [13], Karmakar and Sahoo [14], Xu and Ye [17], Chakraborty, Saha and Pal [18], and Chen and Huang [19]. The coefficient reduced to 5 when $l = 2$, or $\frac{1}{l-2}$ when $l \geq 3$.*

The deficient function is an important definition in the value distribution theory. It is a generalization of the deficiency. It is natural to estimate the deficient small function $a(z)$ with respect to $f^l(f^{(k)})^n(z)$. We obtain the following result, which improves Corollary 1.1 in [19].

**Corollary 1.** *Let $f$ be a transcendental meromorphic function and $\alpha(\not\equiv 0)$ be a small function of $f$. $l(\geq 2), n(\geq 1), k(\geq 1)$ are integers. Then,*

$$\Theta(\alpha, f^l(f^{(k)})^n) \leq 1 - \frac{1}{M(nk+n+l)},$$

*for $M = \min\{\dfrac{1}{l-2}, 5\}$.*

**Remark 2.** *Let $\alpha(\not\equiv 0)$ be a small function of $f$, then we have $\delta(\alpha, f^l(f^{(k)})^n) < 1$. From this, we can obtain a Picard-type theorem. If $f(z)$ is a transcendental meromorphic function and $\alpha(\not\equiv 0)$ is a small function of $f$, then $f^l(f^{(k)})^n - \alpha = 0$ has infinite solutions. In 1939, Titchmarsh [20] considered the differential equation $ff' = -\sin z$ and obtained the solution $f = \pm\sin z$. Many authors consider the nonlinear differential equation including the differential polynomial $f^n f^{(k)}$. For example, Zhang and Yi [21] studied the differential equation $f(z)f'(z) = \frac{1}{2}\sin 2z$, and obtained the solutions of the equation as $f(z) = \pm\sin z, \pm i\cos z$. They also consider the corresponding perturbed equation $f(z)f'(z) = \frac{1}{2}\sin 2z + p(z)$, where $p(z) \not\equiv 0$ is a polynomial, and proved that the equation does not possess an entire solution. In fact, the two differential equation include the*

*differential polynomial $ff'$. Many differential equation can be considered by using the differential polynomial to instead of the derivative (see [22–26]).*

This paper is organized as follows. The lemmas will be used for the proofs of Theorems 2 and 3 in Section 2. The proof of Theorem 2 is placed in Section 3, and an application to the sum of deficiency function is in Section 4. At last, we give a conclusion in Section 5.

## 2. Lemmas

In order to prove our results, we need the following lemmas.

**Lemma 1** ([27]). *Let $f$ be a non-constant meromorphic function, and let $M_1[f], M_2[f]$ be two quasi-differential polynomials in $f$, satisfying $f^n M_1[f] = M_2[f]$. If the total degree of $M_2[f]$ is inferior or equal to $n$, then*

$$m(r, M_1[f]) = S(r, f).$$

**Lemma 2** ([19]). *Let $f$ be a transcendental meromorphic function and $\varphi(\not\equiv 0)$ be a small function of $f$. Then, $\varphi f^l (f^{(k)})^n$ is not equivalent to a constant, where $l(\geq 2), n(\geq 1), k(\geq 1)$ are integers.*

**Lemma 3** ([19]). *Let $f$ be a transcendental meromorphic function, and let $\varphi(z)(\not\equiv 0)$ be a small function of $f$, and a be a non-zero constant. Suppose that $H = \varphi f^l (f^{(k)})^n - a$, where $l(\geq 2)$, $n(\geq 1), k(\geq 1)$ are integers. Then,*

$$
\begin{aligned}
(n+l)T(r,f) \quad &\leq \overline{N}(r,f) + \overline{N}(r, \frac{1}{f}) + nN_{k)}(r, \frac{1}{f}) + nk\overline{N}_{(k+1}(r, \frac{1}{f}) \\
&+ \overline{N}(r, \frac{1}{H}) - N_0(r, \frac{1}{H'}) + S(r,f),
\end{aligned}
\tag{1}
$$

*where $N_0(r, \frac{1}{H'})$ denotes the counting function of the zeros of $H'$, which are not zeros of $f$ or $H$.*

**Remark 3.** *When $l = 2$, the above lemmas have been proved by Chen and Huang in [19]. When $l \geq 3$, we can obtain the results in the similar way (see also [13,15,17,28,29]).*

**Lemma 4** ([2]). *Let $f$ be a transcendental meromorphic function and $b_i, i = 0, 1, \ldots, n$ be small functions of $f$. If*

$$b_n f^n + b_{n-1} f^{n-1} + \cdots + b_0 \equiv 0,$$

*then $b_i \equiv 0, i = 0, 1, \ldots n$.*

In the following, we will give some notations for the next lemmas.

Suppose that $H(z) = \varphi f^2 (f^{(k)})^n - a$ and

$$
h(z) = \frac{H'(z)}{f(z)} = \varphi' f(f^{(k)})^n + 2\varphi f'(f^{(k)})^n + n\varphi f(f^{(k)})^{n-1} f^{(k+1)}, \quad \phi(z) = \frac{h(z)}{H(z)},
$$

where $n(\geq 1), k(\geq 1)$ are integers. Also, let

$$
\begin{aligned}
G(z) \quad &= a_1 \left(\frac{H'(z)}{H(z)}\right)^2 + a_2 \left(\frac{H'(z)}{H(z)}\right)' + a_3 \left(\frac{h'(z)}{h(z)}\right)^2 + a_4 \left(\frac{h'(z)}{h(z)}\right)' + a_5 \left(\frac{H'(z)}{H(z)} \frac{h'(z)}{h(z)}\right) \\
&+ a_6 \left(\frac{\varphi'(z)}{\varphi(z)}\right)^2 + a_7 \left(\frac{\varphi'(z)}{\varphi(z)}\right)' + a_8 \left(\frac{H'(z)}{H(z)} \frac{\varphi'(z)}{\varphi(z)}\right) + a_9 \left(\frac{h'(z)}{h(z)} \frac{\varphi'(z)}{\varphi(z)}\right),
\end{aligned}
$$

where $a_i's$ are defined by

$$\begin{cases} a_1 = -2n^5 - 4n^4 + 5n^3 + 10n^2 + 8n; \\ a_2 = -2(n+1)(15n^4 + 72n^3 + 100n^2 + 48n); \\ a_3 = -2(n+1)^2(n^3 + 8n^2 + 14n - 4); \\ a_4 = 2(n+1)(15n^4 + 73n^3 + 94n^2 + 28n - 8); \\ a_5 = 4(n+1)(n^4 + 5n^3 + 9n^2 + 6n); \\ a_6 = -(36n^4 + 3n^3 - 96n^2 + 101n - 42); \\ a_7 = 4(n+1)(-2n^4 + n^3 + 27n^2 + 20n - 4); \\ a_8 = -2(n+1)(7n^3 + 16n^2 + 10n + 4); \\ a_9 = 2(n+1)(9n^3 + 13n^2 - 14n + 8), \end{cases}$$

when $k = 1$, and are defined by

$$\begin{cases} a_1 = 3b^5 - b^4 - 58b^3 + 12b^2 - 40b - 32; \\ a_2 = -b^6 + 15b^5 - 42b^4 - 436b^3 - 40b^2 - 32b; \\ a_3 = 4b(b-2)(b^2 + 2b); \\ a_4 = 2b^2(b-2)(b^3 - 5b^2 - 4); \\ a_5 = 4b(b^4 + b^3 - 20b^2 - 12b - 16); \\ a_6 = (b-2)(6b^5 - 19b^4 - 61b^3 + 58b^2 - 48b + 64); \\ a_7 = b(b-2)(-b^4 + b^3 + 20b); \\ a_8 = -2b(5b^4 - 37b^3 + 88b^2 - 36b + 32); \\ a_9 = 4b(b-2)(3b^3 - 13b^2 + 8b - 4), \end{cases}$$

when $k \geq 2$, where $b = nk + n + 2$.

We define $\omega(f, z_0) = l$, $\overline{\omega}(f, z_0) = 1$ if $z_0$ is a pole of $f(z)$ with multiplicity $l$. Otherwise, $\omega(f, z_0) = 0$, $\overline{\omega}(f, z_0) = 0$.

**Lemma 5** ([17], Lemma 4). *Under the hypothesis of Theorem 2 and supposing that* $h(z) = \dfrac{H'}{f}$, *for any* $z_0 \in \mathbb{C}$, *we have*

$$\omega(\frac{1}{f}, z_0) + \omega(\frac{1}{h}, z_0) \leq \omega(\frac{1}{fh}, z_0) + \omega(\varphi, z_0) + \omega(\frac{1}{\varphi}, z_0). \tag{2}$$

**Lemma 6** ([17], Lemma 5). *Under the hypotheses of Theorem 2, if* $z_0 \in \mathbb{C}$ *and* $G(z_0) = 0$, *then*

$$\omega(\phi, z_0) \leq 2\omega(\varphi, z_0) + \omega(\frac{1}{\varphi}, z_0), \tag{3}$$

$$\omega(\frac{1}{H'}, z_0) \leq \omega(\frac{1}{h}, z_0) + 2\omega(\varphi, z_0) + \omega(\frac{1}{\varphi}, z_0). \tag{4}$$

**Lemma 7.** *Let* $f$ *be a transcendental meromorphic function, and let* $\varphi(z)(\not\equiv 0)$ *be a small function of* $f$, *where* $a$ *is a non-zero constant. Then,* $G(z) \not\equiv 0$.

**Proof.** Suppose that $G(z) \equiv 0$; then, from Lemma 6, we have

$$N(r, \infty; \phi) \leq 2N(r, \infty; \varphi) + N(r, 0; \varphi) = S(r, f), \tag{5}$$

and

$$\begin{aligned} N(r, 0; H) &\leq 2N(r, \infty; \varphi) + N(r, 0; \varphi) + N(r, 0; h) \\ &= N(r, 0; h) + S(r, f). \end{aligned} \tag{6}$$

It follows from Lemma 2 that $H$ is not identically constant. Let

$$\frac{a}{f^{n+2}} \equiv \varphi(\frac{f^{(k)}}{f})^n - \frac{H'}{f^{n+2}}\frac{H}{H'}.$$

By the lemma of the logarithmic derivative, we have

$$
\begin{aligned}
(n+2)m(r,\frac{1}{f}) &\leq m(r,\varphi) + m(r,(\frac{f^{(k)}}{f})^n) + m(r,\frac{H'}{f^{n+2}}) + m(r,\frac{H}{H'}) \\
&= N(r,\infty;\frac{H'}{H}) - N(r,\infty;\frac{H}{H'}) + S(r,f) \\
&= \overline{N}(r,\infty;f) + N(r,0;H) - N(r,0;H') + S(r,f).
\end{aligned}
\tag{7}
$$

From (7), we have

$$(n+2)m(r,\frac{1}{f}) \leq \overline{N}(r,\infty;f) + N(r,0;H) - N(r,0;hf) + S(r,f). \tag{8}$$

From Lemma 5, we have

$$N(r,0;f) + N(r,0;h) \leq N(r,0;hf) + N(r,\infty;\varphi) + N(r,0;\varphi). \tag{9}$$

From (8) and (9), we have

$$(n+2)m(r,\frac{1}{f}) + N(r,0;f) \leq \overline{N}(r,\infty;f) + N(r,0;H) - N(r,0;h) + S(r,f). \tag{10}$$

From (6) and (10), we have

$$(n+1)m(r,\frac{1}{f}) \leq N(r,0;H) - N(r,0;h) + S(r,f) = S(r,f). \tag{11}$$

From (5) and (11), we have

$$
\begin{aligned}
T(r,\phi) &= m(r,\phi) + N(r,\infty;\phi) = m(r,\frac{1}{f}\frac{H'}{H}) + N(r,\infty;\phi) \\
&\leq m(r,\frac{1}{f}) + m(r,\frac{H'}{H}) + S(r,f) = S(r,f).
\end{aligned}
\tag{12}
$$

Note that

$$\frac{H'}{H} = f\phi$$

and

$$\frac{h'}{h} = \frac{H'}{H} + \frac{\phi'}{\phi} = f\phi + \frac{\phi'}{\phi}.$$

Substituting the above two equalities into $G(z)$ yields

$$
\begin{aligned}
&(a_1 + a_3 + a_5)f^2\phi^2 + (a_2 + a_4)f'\phi + [(a_2 + 2a_3 + a_4 + a_5)\frac{\phi'}{\phi} + (a_8 + a_9)\frac{\varphi'}{\varphi}]f\phi \\
&+ (a_3(\frac{\phi'}{\phi})^2 + a_4(\frac{\phi'}{\phi})' + a_6(\frac{\varphi'}{\varphi})^2 + a_7(\frac{\varphi'}{\varphi})' + a_9(\frac{\phi'}{\phi}\frac{\varphi'}{\varphi}) \equiv 0.
\end{aligned}
\tag{13}
$$

If $k = 1$, we have $a_2 + a_4 = 2n^4 - 10n^3 - 52n^2 - 56n - 16$. If

$$2n^4 - 10n^3 - 52n^2 - 56n - 16 = 0,$$

then $n = -2, -1, 4 \pm \sqrt{5}$. Note that $n \geq 2$ is a positive integer. Therefore, $a_2 + a_4 \neq 0$.

If $k \geq 2$, we have $a_2 + a_4 = b^5 + 10b^4 + 36b^3 + 56b^2 + 32b$. Noting that $b = nk + n + 2 \geq 6$, we immediately obtain $a_2 + a_4 \neq 0$.

Obviously, $\phi \not\equiv 0$, otherwise $\dfrac{H'}{H} = f\phi \equiv 0$; that is, $H \equiv C$. This contradicts Lemma 2. Hence, we can obtain the following relation from (13):

$$f' = \frac{1}{\phi}c_{11}(z) + fc_{12}(z) + f^2\phi c_{13}(z), \tag{14}$$

where $c_{1i}(i = 1, 2, 3)$ are small functions of $f$. Differentiating both sides of (14) gives

$$f'' = \frac{1}{\phi}c_{21}(z) + fc_{22}(z) + f^2\phi c_{23}(z) + f^3\phi^2 c_{24}(z),$$

where $c_{2i}(i = 1, 2, 3, 4)$ are small functions of $f$. Continuing the above process, we obtain

$$f^{(k)} = \frac{1}{\phi}c_{k1}(z) + fc_{k2}(z) + f^2\phi c_{k3}(z) + \cdots + f^{k+1}\phi^k c_{kk+2}(z), \tag{15}$$

where $c_{ki}(i = 1, 2, \cdots, k + 2)$ are small functions of $f$.

From (15), we have

$$H(z) = \varphi f^2 (f^{(k)})^n - a = \varphi \phi^{nk} c_{kk+2}^n f^{nk+n+2} + \cdots + \varphi f^2 \frac{1}{\phi^n} c_{k1}^n - a.$$

Let us take the derivative of above equality; from the equation $H' = f\phi F$, the coefficient of $f$ in $H' - f\phi H$ is $a\phi$. By Lemma 4, we have $a\phi \equiv 0$. Notice that $a \neq 0$ and $\phi \not\equiv 0$, which is a contradiction. Hence, $G(z) \not\equiv 0$.

This completes the proof of Lemma 7.  □

**Lemma 8.** *Let $f(z), H(z), h(z)$ and $G(z)$ be stated as the above. Then, all simple poles of $f(z)$ are the zeros of $G(z)$.*

**Proof.** Suppose that $z_0$ is a simple pole of $f(z)$, then

$$\varphi(z) = B\{1 + x(z - z_0) + y(z - z_0)^2 + O((z - z_0)^3)\},$$
$$f(z) = \frac{A}{z - z_0}\{1 + c_0(z - z_0) + c_1(z - z_0)^2 + O((z - z_0)^3)\},$$

where $AB \neq 0, x, y, c_0, c_1$ are constants. Next, we consider two cases.

**Case 1**. $k = 1$. We have

$$\begin{aligned}
H(z) = \varphi f^2 (f')^n - a &= \frac{(-1)^n A^{n+2} B}{(z - z_0)^{2n+2}}\{1 + (2c_0 + x)(z - z_0) \\
&\quad + [c_0^2 + y + 2c_0 x + (2 - n)c_1](z - z_0)^2 + O((z - z_0)^3)\}, \\
h(z) = \frac{F'(z)}{f(z)} &= \frac{(-A)^{n+1} B}{(z - z_0)^{2n+2}}\{2n + 2 + [2nc_0 + (2n - 1)x](z - z_0) \\
&\quad + [2ny + (2n - 1)c_0 x - 2n(n - 2)c_1](z - z_0)^2 + O((z - z_0)^3)\}.
\end{aligned}$$

Therefore, we have

$$\begin{aligned}
\frac{H'(z)}{H(z)} &= \frac{-1}{z - z_0}\{2n + 2 - (2c_0 + x)(z - z_0) \\
&\quad + [2c_0^2 + x^2 - 2y + (2n - 4)c_1](z - z_0)^2 + O((z - z_0)^3)\},
\end{aligned}$$

$$\frac{h'(z)}{h(z)} = \frac{-1}{z - z_0}\{2n + 2 - \frac{2nc_0 + (2n-1)x}{2n+2}(z - z_0)$$
$$+ \frac{1}{2n+2}[\frac{4c_0x + 4n^2c_0^2 + (2n-1)x^2}{2n+2} + 4n(n-2)c_1 - 4ny](z - z_0)^2$$
$$+ O((z - z_0)^3)\},$$

$$\frac{\varphi'(z)}{\varphi(z)} = x + (2y - x^2)(z - z_0) + O((z - z_0)^2),$$

$$\left(\frac{H'(z)}{H(z)}\right)^2 = \frac{1}{(z - z_0)^2}\{(2n+2)^2 - (4n+4)(2c_0 + x)(z - z_0)$$
$$+ [(8n+12)c_0^2 + (4n+5)x^2 + 4c_0x + (4n+4)(2n-4)c_1$$
$$- (8n+8)y](z - z_0)^2 + O((z - z_0)^3)\},$$

$$\left(\frac{H'(z)}{H(z)}\right)' = \frac{1}{(z - z_0)^2}\{2n + 2 + [2c_0^2 + x^2 + (2n-4)c_1 - 2y](z - z_0)^2$$
$$+ O((z - z_0)^3)\},$$

$$\left(\frac{h'(z)}{h(z)}\right)^2 = \frac{1}{(z - z_0)^2}\{(2n+2)^2 - [4nc_0 + (4n-2)x](z - z_0)$$
$$+ [\frac{4n^2(4n+5)c_0^2 + (4n+5)(2n-1)^2x^2 + (8n^2+12n+16)c_0x}{(2n+2)^2}$$
$$+ 8n(n-2)c_1 - 8ny](z - z_0)^2 + O((z - z_0)^3)\},$$

$$\left(\frac{h'(z)}{h(z)}\right)' = \frac{1}{(z - z_0)^2}\{2n + 2 + \frac{1}{2n+2}[\frac{4n^2c_0^2 + (2n-1)^2x^2 + 4c_0x}{2n+2}$$
$$+ 4n(n-2)c_1 - 4ny](z - z_0)^2 + O((z - z_0)^3)\},$$

$$\frac{H'(z)}{H(z)}\frac{h'(z)}{h(z)} = \frac{1}{(z - z_0)^2}\{(2n+2)^2 - [(6n-4)c_0 + (4n-1)x](z - z_0)$$
$$+ [(6n+4)c_0^2 + \frac{(8n^2+6n+4)x^2 + (6n+2)c_0x}{2n+2} + (2n-4)(4n+2)c_1$$
$$- (8n+4)y](z - z_0)^2 + O((z - z_0)^3)\},$$

$$\left(\frac{\varphi'(z)}{\varphi(z)}\right)^2 = x^2 + 2x(2y - x^2)(z - z_0) + O((z - z_0)^2),$$

$$\left(\frac{\varphi'(z)}{\varphi(z)}\right)' = 2y - x^2 + O(z - z_0),$$

$$\frac{H'(z)}{H(z)}\frac{\varphi'(z)}{\varphi(z)} = \frac{-1}{z - z_0}\{(2n+2)x + [(4n+4)y - 2c_0x - (2n+1)x^2](z - z_0)$$
$$+ O((z - z_0)^2)\},$$

$$\frac{h'(z)}{h(z)}\frac{\varphi'(z)}{\varphi(z)} = \frac{-1}{z - z_0}\{(2n + 2)x + [(4n + 4)y - \frac{2n}{2n + 2}c_0 x$$

$$- \frac{4n^2 + 10n - 3}{2n + 2}x^2](z - z_0) + O((z - z_0)^2)\},$$

Putting the above equalities into $G(z)$ and making some easy calculations, we have $G(z) = O(z - z_0)$, and $z_0$ is a zero of $G(z)$.

**Case 2**. $k \geq 2$. We have

$$H(z) = \varphi f^2(f^{(k)})^n - a = \frac{(-1)^{nk}(k!)^n A^{n+2}B}{(z - z_0)^{nk+n+2}}\{1 + (2c_0 + x)(z - z_0)$$

$$+ (c_0^2 + 2c_0 x + 2c_1 + y)(z - z_0)^2 + O((z - z_0)^3)\},$$

$$h(z) = \frac{(-1)^{nk+1}(k!)^n A^{n+1}B}{(z - z_0)^{nk+n+2}}\{nk + n + 2 + [n(k+1)c_0 + (nk + n + 1)x](z - z_0)$$

$$+ [(nk + n - 1)c_0 x + (nk + n - 2)c_1 + (nk + n)y](z - z_0)^2 + O((z - z_0)^3)\}.$$

Using the two above equalities, we obtain

$$\frac{H'(z)}{H(z)} = \frac{-1}{z - z_0}\{nk + n + 2 - (2c_0 + x)(z - z_0)$$

$$+ [2c_0^2 + x^2 - 4c_1 - 2y](z - z_0)^2 + O((z - z_0)^3)\},$$

$$\frac{h'(z)}{h(z)} = \frac{-1}{z - z_0}\{nk + n + 2 - \frac{(nk + n)c_0 + (nk + n + 1)x}{nk + n + 2}(z - z_0)$$

$$+ [\frac{(nk + n)^2 c_0^2 + (nk + n + 1)x + 4c_0 x}{(nk + n + 2)^2} - \frac{2(nk + n - 2)}{nk + n + 2}c_1$$

$$- \frac{2(nk + n)}{nk + n + 2}y](z - z_0)^2 + O((z - z_0)^3)\},$$

$$(\frac{H'(z)}{H(z)})^2 = \frac{1}{(z - z_0)^2}\{(nk + n + 2)^2 - 2(nk + n + 2)(2c_0 + x)(z - z_0)$$

$$+ [4(nk + n + 3)c_0^2 + (2nk + 2n + 5)x^2 + 4c_0 x$$

$$- 4(nk + n + 2)(y + 2c_1)](z - z_0)^2 + O((z - z_0)^3)\},$$

$$(\frac{H'(z)}{H(z)})' = \frac{1}{(z - z_0)^2}\{nk + n + 2 - [2c_0^2 + x^2 - 4c_1 - 2y](z - z_0)^2 + O((z - z_0)^3)\},$$

$$(\frac{h'(z)}{h(z)})^2 = \frac{1}{(z - z_0)^2}\{(nk + n + 2)^2 - [2(nk + n)c_0 + (nk + n + 1)x](z - z_0)$$

$$+ [\frac{(2nk + 2n + 5)(nk + n)^2 c_0^2 + 2(nk + n)(nk + n + 1)x^2}{(nk + n + 2)^2}$$

$$+ (\frac{nk + n + 1}{nk + n + 2})^2 c_0 x + \frac{(nk + n + 1)x^2 + 4c_0 x}{nk + n + 2} - 4(nk + n - 2)c_1$$

$$- 2(nk + n)y](z - z_0)^2 + O((z - z_0)^3)\},$$

$$\left(\frac{h'(z)}{h(z)}\right)' = \frac{1}{(z-z_0)^2}\{nk+n+2 - [\frac{(nk+n)^2c_0^2 + (nk+n+1)^2x^2 + 4c_0x}{(nk+n+2)^2}$$
$$- \frac{2(nk+n-2)c_1 + 2(nk+n)y}{nk+n+2}](z-z_0)^2 + O((z-z_0)^3)\},$$

$$\frac{H'(z)}{H(z)}\frac{h'(z)}{h(z)} = \frac{1}{(z-z_0)^2}\{(nk+n+2)^2 - [(3nk+3n+4)c_0 + (2nk+2n+3)x](z-z_0)$$
$$+ [(3nk+3n+4)c_0^2 + (nk+n+2+\frac{2(nk+n+1)}{nk+n+2})x^2 + 3c_0x$$
$$- 2(3nk+3n+2)c_1 - 4(nk+n+1)y](z-z_0)^2 + O((z-z_0)^3)\},$$

$$\left(\frac{\varphi'(z)}{\varphi(z)}\right)^2 = x^2 + 2x(2y - x^2)(z-z_0) + O((z-z_0)^2),$$

$$\left(\frac{\varphi'(z)}{\varphi(z)}\right)' = 2y - x^2 + O(z-z_0),$$

$$\frac{H'(z)}{H(z)}\frac{\varphi'(z)}{\varphi(z)} = \frac{-1}{z-z_0}\{(nk+n+2)x$$
$$+ [(2nk+2n+4)y - 2c_0x - (nk+n+3)x^2](z-z_0) + O((z-z_0)^2)\},$$

$$\frac{h'(z)}{h(z)}\frac{\varphi'(z)}{\varphi(z)} = \frac{-1}{z-z_0}\{(nk+n+2)x + [(2nk+2n+4)y - \frac{nk+n}{nk+n+2}c_0x$$
$$- \frac{(nk+n+2)^2 + nk+n+1}{nk+n+2}x^2](z-z_0) + O((z-z_0)^2)\}.$$

Putting the above equalities into $G(z)$, and making some easy calculations, we again obtain $G(z) = O(z-z_0)$. Hence, the simple pole is the zero of $G(z)$. $\square$

### 3. The Proof of Theorem 2

**Proof of Theorem 2.**

When $l = 2, n \geq 1, k \geq 1$, we consider two cases.

**Case 1**. First, we suppose that $k \geq 2$. From Lemmas 7 and 8, we have $G \not\equiv 0$ and the simple pole of $f(z)$ is the zero of $G(z)$. Set

$$\beta = \varphi'f(f^{(k)})^n + 2\varphi f'(f^{(k)})^n + n\varphi f(f^{(k)})^{n-1}f^{(k+1)} - \varphi f(f^{(k)})^n\frac{H'}{H}.$$

Then, $f\beta = -a\frac{H'}{H}$ and $h = -\frac{1}{a}\beta H$. For $G(z)$, we notice that the poles of $G(z)$ with multiplicities are two at most, which come from the multiple poles of $f(z)$, or from the zeros of $H(z)$, or from the zeros of $h(z)$, or from the zeros of $\varphi(z)$. Since $\varphi(z)$ is a small function of $f(z)$, we ignore its zeros and poles here.

Now, we consider the poles of $\beta G$. The zeros of $h$ are either the zeros of $H$ or the zeros of $\beta$. From the above discussion, we can find that the multiple poles of $f$ with multiplicity $q(\geq 2)$ are the zeros of $\beta$ with multiplicity of $q - 1$. Hence, the poles of $\beta G$ only come from the zeros of $F$ and the multiplicity is at most 3. Thus,

$$N(r, \infty; \beta G) \leq 3\overline{N}(r, 0; H).$$

Noting $m(r, G) = S(r, f)$ and $m(r, \beta) = S(r, f)$ from Lemma 1, we have

$$m(r, \beta G) = S(r, f).$$

Therefore,

$$T(r, \beta G) \leq 3\overline{N}(r, 0; H) + S(r, f).$$

Since the simple poles of $f$ are the zeros of $\beta G$, we obtain

$$N_1(r, \infty; f) \leq N(r, 0; \beta G) \leq T(r, \beta G) \leq 3\overline{N}(r, 0; H).$$

It follows from the above equality and double (1) that

$$2(n+2)T(r, f) + N_1(r, \infty; f) + \leq 2\overline{N}(r, \infty; f) + 2\overline{N}(r, 0; f) + 5\overline{N}(r, 0; H) + 2nN_{k)}(r, 0; f)$$
$$+ 2nk\overline{N}_{(k+1}(r, 0; f) - 2N_0(r, 0; H') + S(r, f),$$

which leads to

$$T(r, f) + 2(n+1)m\left(r, \frac{1}{f}\right) + m(r, f) + [N(r, \infty; f) + N_1(r, \infty; f) - 2\overline{N}(r, \infty; f)]$$
$$+ 2[N(r, 0; f) - \overline{N}(r, 0; f)] + 2n[N(r, 0; f) - N_{k)}(r, 0; f) - k\overline{N}_{(k+1}(r, 0; f)] \qquad (16)$$
$$\leq 5\overline{N}(r, 0; H) - N_0(r, 0; H') + S(r, f).$$

Note that

$$N(r, \infty; f) + N_1(r, \infty; f) - 2\overline{N}(r, \infty; f) \geq 0, \quad N(r, 0; f) - \overline{N}(r, 0; f) \geq 0,$$

and

$$N(r, 0; f) - N_{k)}(r, 0; f) - k\overline{N}_{(k+1}(r, 0; f) = N_{(k+1}(r, 0; f) - k\overline{N}_{(k+1}(r, 0; f) \geq 0,$$

so we immediately obtain

$$T(r, f) \leq 5\overline{N}(r, 0; H) + S(r, f).$$

**Case 2.** Suppose that $k = 1$. Set

$$\beta = \varphi' f(f')^n + 2\varphi(f')^{n+1} + n\varphi f(f')^{n-1} f'' - \varphi f(f')^n \frac{H'}{H}.$$

Similarly to Case 1, we obtain the same conclusion.

When $l \geq 3$, from Lemma 3, we have

$$(l-2)T(r, f) + (n+1)m\left(r, \frac{1}{f}\right) + m(r, f) + [N(r, \infty; f) - \overline{N}(r, \infty; f)]$$
$$+ [N(r, 0; f) - \overline{N}(r, 0; f)] + n[N(r, 0; f) - N_{k)}(r, 0; f) - k\overline{N}_{(k+1}(r, 0; f)]$$
$$\leq \overline{N}(r, 0; H) - N_0(r, 0; H') + S(r, f).$$

Note that

$$N(r, \infty; f) - \overline{N}(r, \infty; f) \geq 0, \quad N(r, 0; f) - \overline{N}(r, 0; f) \geq 0,$$

and

$$N(r, 0; f) - N_{k)}(r, 0; f) - k\overline{N}_{(k+1}(r, 0; f) = N_{(k+1}(r, 0; f) - k\overline{N}_{(k+1}(r, 0; f) \geq 0,$$

so we immediately obtain

$$T(r, f) \leq \frac{1}{l-2}\overline{N}(r, 0; H) + S(r, f).$$

This completes the proof of Theorem 2.   □

## 4. An Application

It is well-known that, for the nonconstant meromorphic function $f$, $\sum_{a \in \mathbb{C}} \delta(a, f) \leq 2$ at most countable many deficient values of $f$. For the function $f^{(k)}$, Mues [5] posed the conjecture: $\sum_{a \in \mathbb{C}} \delta(a, f) \leq 1$ $(k \geq 1)$. Yamanoi [30] confirmed Mues conjecture for $k = 1$. Fang and Wang [31] considered for any $k \geq 1$, and obtained the result

$$\sum_{a \in \mathbb{C}} \delta\left(a, f^{(k)}\right) + \sum_{j=k+1}^{\infty} \sum_{b \in \mathbb{C}/\{0\}} \delta\left(b, f^{(j)}\right) \leq 1.$$

For the differential polynomial, it is natural to consider the deficiency relations of the differential polynomial. Jiang and Huang [32] gave a result for $f^l \left( f^{(k)} \right)^n - a$, where $l(\geq 2), n(\geq 2), k(\geq 2)$ and $a \in \mathbb{C}$. In this paper, we improve the result where $l(\geq 2), n(\geq 1), k(\geq 1)$ and $a$ is a small function of $f$.

**Theorem 3.** *Let $f$ be a transcendental meromorphic function in $\mathbb{C}$, $l(\geq 2), n(\geq 1), k(\geq 1)$ be positive integers and $a_i$ be small functions of $f$, $i = 1, 2, \ldots, q$. Then,*

$$\sum_{i=1}^{q} \delta\left(a_i, f^l \left( f^{(k)} \right)^n\right) \leq 1 + \frac{1}{nk + n + l}.$$

**Proof.** Let $\phi = \varphi f^l \left( f^{(k)} \right)^n$. By Lemma 1.7 in [2], we have

$$\sum_{i=1}^{q} m\left(r, \frac{1}{\phi - b_i}\right) = m\left(r, \sum_{i=1}^{q} \frac{1}{\phi - b_i}\right) + O(1)$$

$$\leq m\left(r, \sum_{i=1}^{q} \frac{\phi''}{\phi - b_i}\right) + m\left(r, \frac{1}{\phi''}\right) + S(r, f)$$

$$\leq T\left(r, \phi''\right) - N\left(r, 0; \phi''\right) + S(r, f)$$

$$\leq N\left(r, \infty; \phi''\right) + m\left(r, \phi''\right) - N\left(r, 0; \phi''\right) + S(r, f)$$

$$\leq N\left(r, \infty; \phi\right) + 2\bar{N}\left(r, \infty; \phi\right) + m\left(r, \phi\right) - N\left(r, 0; \phi''\right) + S(r, f).$$

By Lemma 1 in [31], we see that

$$\sum_{i=1}^{q} m\left(r, \frac{1}{\phi - b_i}\right) \leq T(r, \phi) + 2\bar{N}(r, \infty; \phi) - \bar{N}(r, \infty; \phi) + S(r, f)$$

$$\leq T(r, \phi) + \bar{N}(r, \infty; f) + S(r, f)$$

$$\leq T(r, \phi) + T(r, f) + S(r, f).$$

Let $a_i = b_i / \varphi (i = 1, \ldots, q)$. Note that

$$N(r, b_i; \phi) = N\left(r, a_i; f^l \left( f^{(k)} \right)^n\right) + S(r, f).$$

By Theorem 2 and Corollary 1, it follows from inequality (22) that

$$
\begin{aligned}
\sum_{i=1}^{q} \delta\left(a_i, f^l\left(f^{(k)}\right)^n\right) &= \liminf_{r \to \infty} \sum_{i=1}^{q} \frac{m\left(r, \frac{1}{\phi - b_i}\right)}{T(r, \phi)} \\
&\leq 1 + \liminf_{r \to \infty} \frac{T(r, f)}{T(r, \phi)} + S(r, f) \\
&\leq 1 - \frac{1}{l - 2}\left(1 - \limsup_{r \to \infty} \frac{\overline{N}(r, a; \phi)}{T(r, \phi)} - 1\right) \\
&= 1 - \frac{1}{l - 2}(\delta(a, \phi) - 1) \\
&\leq 1 - \frac{1}{l - 2}\left(1 - \frac{l - 1}{nk + n + l} - 1\right) \\
&= 1 + \frac{1}{nk + n + l}
\end{aligned}
$$

This completes the proof of Theorem 3. $\square$

## 5. Conclusions

In this paper, we mainly consider the estimation of the characteristic function by the reduced counting function. In the Nevanlinna theory, the second main theorem is the most important theorem, where the characteristic function $T(r, f)$ is bounded by three reduced counting functions. Also, we know that the characteristic function $T(r, f)$ is bounded by two counting functions, considering the derivative of $f$ in Hayman's inequality, but the coefficients of the two counting functions seems too large, not as excepted as those ones which equal 1 in Nevanlinna's second function main theorem. For this sake, Hayman [1] asked whether or not the coefficients of $N(r, 0; f)$ and $N(r, 1; f^{(k)})$ in the inequality are best. L. Yang [2] answered this question, and gave a small coefficient $1 + \frac{1}{k}$. Recently, Fang and Wang [31] obtained a more precise coefficient $\frac{1}{1-\varepsilon}$, and one counting function $N(r, 0; f)$ is replaced by the reduced function $\overline{N}(r, 0; f)$ using the result of Yamanoi [30]. In this direction, it is natural to study the characteristic function $T(r, f)$, which is bounded by one counting function, considering the product of a meromorphic function $f$ and its derivative (differential monomials or differential polynomials). Yi [33] give a quantitative estimation for the differential polynomial $f^l(f^{(k)})^n - a(l \geq 3, n, k \geq 1)$ using the reduced counting function; the coefficients is $\frac{1}{n-2}$. Lahiri and Dewan [34] also obtained a similar result using the counting function. For the case $l = 2$, Zhang [10] and Huang and Gu [11] determined that a quantitative result $T(r, f) < 6N(r, 1; f^2 f^{(k)}) + S(r, f)$ holds for $k = 1$ and $k \geq 2$, respectively. Jiang and Huang [32] also obtained an estimate for $f^l(f^{(k)})^n - a(l, n, k \geq 2)$ by counting function using the result of Yamanoi [30]; the coefficients is $\frac{1}{n-1}$. That is, $T(r, f) \leq \frac{1}{n-1} N(r, a; f^l(f^{(k)})^n) + S(r, f)$. However, the key of their proof is the result of Yamanoi, which is that the counting function cannot be replaced by the reduced counting function. Our results mainly consider the quantitative estimation for the differential polynomial using the reduced counting function. They improve and generalize the existing literature ([14,19,28,35–40]).

**Author Contributions:** Writing Original Draft Preparation, J.L. and J.X.; Writing Review and Editing, J.X. All authors have read and agreed to the published version of the manuscript.

**Funding:** This research was supported by the National Natural Science Foundation of China (no.11871379), Natural Science Foundation of Guangdong (no.2021A1515010062).

**Data Availability Statement:** No new data were created or analyzed in this study. Data sharing is not applicable to this article.

**Conflicts of Interest:** The authors declare no conflicts of interest.

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
