# Peer review of "On the Zeros of the Differential Polynomials"

_mathematics, doi:10.3390/math12081196_

Round 1
Reviewer 1 Report
Comments and Suggestions for Authors
see attachment

it is ok
Author Response
We are grateful to Reviewer 1 for your helpful comments
and suggestions. Now we have made modifications following these
suggestions completely.

Reviewer 2 Report
Comments and Suggestions for Authors
Author Response
We are grateful to Reviewer 2 for your helpful comments
and suggestions. Now we have made modifications following these
suggestions completely.

Reviewer 3 Report
Comments and Suggestions for Authors
The paper evaluates the characteristic function using the reduced counting function for differential polynomial $\varphi f^l(f^{(k)})^n-a$. There are some issues that the authors need to consider:
1. It is necessary to explain Remark 1 in more detail (it is not obvious).
2. After stating Theorem 1.1, you need to provide a plan for its proof.
3. An explanation is needed after stating Corollary 1.2 in the first section that its proof is given in the fourth section.
4. There is no proofs of Lemma 2.5, Lemma 2.6 (analogy does not mean complete coincidence, details are needed).
5. The iThenticate analysis revealed a text similarity of over 50%, indicating that it is not suitable for an academic report or research paper.
Author Response
We are grateful to Reviewer 3 for your helpful comments
and suggestions. Now we have made modifications following these
suggestions completely.

Reviewer 4 Report
Comments and Suggestions for Authors
I read throughout the manuscript again so that I can evaluate the research work objectively and accurately.
(1)At the start, I spot some grammar errors, specifically, in the 5th line of Abstract, “Our results improves and generalizes the results ...” should be “Our results improve and generalize the results...”. Hence, I suggest authors to check the manuscript carefully so it is free of grammar errors.
(2)I can not say something about the significance of the results stated in Theorem 1.1 and Corollary 1.2.
(3)The research work belongs to pure mathematics, and I think the results may be evaluated in some days. The manuscript does not include any tables and figures.
So, I think that the manuscript can be considered for publishing after minor revision.
1. What is the main question addressed by the research? Answer: 2. What parts do you consider original or relevant for the field? What specific gap in the field does the paper address? Answer: An novel result is presented in the research, which is stated in Theorem 1.1. 3. What does it add to the subject area compared with other published material? 4. What specific improvements should the authors consider regarding the methodology? What further controls should be considered? 5. Please describe how the conclusions are or are not consistent with the evidence and arguments presented. Please also indicate if all main questions posed were addressed and by which specific experiments. 6. Are the references appropriate? 7. Please include any additional comments on the tables and figures and quality of the data.

Author Response
We are grateful to Reviewer 4 for your helpful comments
and suggestions. Now we have made modifications following these
suggestions completely.

Round 2
Reviewer 1 Report
Comments and Suggestions for Authors
-
The manuscript has been revised according to my comments, and I recommend to accept it
it is ok
Reviewer 2 Report
Comments and Suggestions for Authors
The author has revised the article. He made the necessary additions to the article.
The article can be published as is.
Reviewer 3 Report
Comments and Suggestions for Authors
I see that the authors have addressed all my comments accordingly. The manuscript is revised and improved. Thus, I recommend this article for publication in the journal Mathematics.